# The Effects of Selenium on Bone Health: From Element to Therapeutics

**DOI:** 10.3390/molecules27020392

**Published:** 2022-01-08

**Authors:** Taeyoung Yang, So-Young Lee, Kyung-Chae Park, Sin-Hyung Park, Jaiwoo Chung, Soonchul Lee

**Affiliations:** 1Department of Internal Medicine, CHA Bundang Medical Center, CHA University School of Medicine, Seongnam-si 13496, Korea; taeyoungy1@chamc.co.kr (T.Y.); ysy0119@cha.ac.kr (S.-Y.L.); 2Health Promotion Center, CHA Bundang Medical Center, CHA University School of Medicine, Seongnam-si 13488, Korea; kc829@cha.ac.kr; 3Department of Orthopaedic Surgery, Bucheon Hospital, Soonchunhyang University School of Medicine, Bucheon-si 14584, Korea; sh0803@schmc.ac.kr; 4Department of Orthopaedic Surgery, CHA Bundang Medical Center, CHA University School of Medicine, Seongnam-si 13496, Korea; jwc.os.cha@gmail.com

**Keywords:** selenium, selenoprotein, bone mineral density, osteoporosis, therapeutics, nanoparticles, bone formation

## Abstract

Osteoporosis, characterized by low bone mass and a disruption of bone microarchitecture, is traditionally treated using drugs or lifestyle modifications. Recently, several preclinical and clinical studies have investigated the effects of selenium on bone health, although the results are controversial. Selenium, an important trace element, is required for selenoprotein synthesis and acts crucially for proper growth and skeletal development. However, the intake of an optimum amount of selenium is critical, as both selenium deficiency and toxicity are hazardous for health. In this review, we have systematically analyzed the existing literature in this field to determine whether dietary or serum selenium concentrations are associated with bone health. In addition, the mode of administration of selenium as a supplement for treating bone disease is important. We have also highlighted the importance of using green-synthesized selenium nanoparticles as therapeutics for bone disease. Novel nanobiotechnology will be a bridgehead for clinical applications of trace elements and natural products.

## 1. Introduction

Bones play multiple roles in the body, including providing support to the body and an environment for generating blood cells, enabling movement, protecting vital organs, and storing minerals [1,2,3]. Thus, it is not surprising that the deterioration of bone health results in morbidity, disability, and occasionally worse outcomes [4,5]. Osteoporosis, which is characterized by low bone mass and a disruption of bone microarchitecture, is the most common bone disease in humans [6,7,8,9,10]. It is prevalent in menopausal women and elderly men and is a major risk factor for fractures [11,12]. With an increase in longevity and societal aging in many parts of the world, osteoporosis is becoming a global epidemic [13,14].

Several primary and secondary preventive strategies for osteoporosis have been recommended, including drugs, diet, and other lifestyle modifications [15,16]. Among dietary elements, most previous studies have focused on calcium and vitamin D supplementation [17,18,19,20,21,22,23,24], which have traditionally been provided as anti-resorptive treatments for every adult at risk; however, fractures can be prevented in their absence, and the effects of these agents on bone density appear to be negligible [25,26,27].

Selenium is a trace mineral and an indispensable component of various enzymes and proteins [28,29,30]. Selenoprotein is synthesized from selenocysteine, an essential amino acid residue important for DNA synthesis, which protects cells from damage and infection [31,32] and is involved in the reproduction and metabolism of thyroid hormones [33,34]. Although selenium is required in insignificant amounts, approximately 1 billion people reportedly have selenium deficiency worldwide [35]. Inadequate selenium intake may result from certain circumstances, in people living in places with suboptimal selenium content in the soil [36,37,38] or in patients with bowel disease [39,40,41] or chronic kidney disease [42,43,44]. Notably, the risk of developing selenium deficiency might increase because of climate change, which is known to reduce selenium content in soil [35].

Selenium has been reported to be important for skeletal development [45]. Recently, narrative review research [46] with several preclinical and clinical studies have investigated the effects of selenium on bone health, although the results are controversial [47,48,49,50]. In this review, we systematically analyzed the existing literature in this field to determine whether dietary or serum selenium concentrations are associated with bone health. The following Pubmed and Google Scholar keywords were used to review the above publications: selenium AND bone, selenium AND osteoporosis, selenoproteins AND bone metabolism, selenium AND bone cells, selenium AND bone in animal models, selenium AND bone in human study, and selenium AND randomized clinical trial. We expect that this review will contribute to the future development of the selenium-based therapeutics.

## 2. Selenium

### 2.1. Selenium Synthesis

The essential role of selenium in living organisms was identified 140 years after its discovery [51]. Selenium was discovered in 1817 by J. Berzelius and J. Gahn, who investigated a reddish brown residue after manufacturing sulfuric acid [52,53]. Selenium belongs to the family of chalcogens, including oxygen, sulfur, tellurium, and polonium and forms several allotropes that interconvert with changes in temperature [54,55].

Selenium performs its physiological functions principally as a constituent of selenoproteins [45]. Dietary sources of selenium uptake include inorganic forms such as selenate and selenite, as well as organic forms like selenocysteine (Sec) and selenomethionine (SeMet) [56,57]. Each of these forms can be metabolized to selenide, which is an intermediate in selenocysteine synthesis [58,59]. Selenocysteine biosynthesis is unique as it requires its own tRNA, named selenocysteine tRNA^[Ser]Sec^ [60]. After selenide (H_2_Se) conversion to seryl-tRNA^[Ser]Sec^ by seryl-tRNA synthetase, seryl-tRNA^[Ser]Sec^ provides the backbone for selenocysteine biosynthesis. Subsequently, phosphoseryl-tRNA kinase transforms seryl-tRNA^[Ser]Sec^ into phosphoseryl-tRNA^[Ser]Sec^, which acts as a substrate for selenocysteine synthase with selenophosphate. Selenophosphate is generated from selenide and ATP by selenophosphate synthase 2. Selenocysteine incorporation into proteins requires a cis-acting selenocysteine insertion sequence (SECIS) element at a specific UGA codon, which usually acts as a stop codon [61]. When a ribosome reaches the UGA codon, SECIS elements act as the factors that dictate the recoding of UGA as selenocysteine [62]. SECIS binding protein 2 forms a complex with ribosomes, SECIS elements, and a Sec-specific translation elongation factor, contributing to the efficient coding of UGA as selenocysteine [62]. As SECIS binding protein 2 is known to be a limiting factor for selenoprotein synthesis, the expression of SECIS binding protein 2 may regulate selenocysteine incorporation and selenoprotein production in cells [63,64]. The structure of Sec-specific translation elongation factor is similar to that of canonical elongation factor 1A, although its domain IV contains a unique COOH-terminal extension interacting with SECIS binding protein 2 and tRNA^[Ser]sec^ [65,66]. In addition, additional SECIS-binding proteins such as ribosomal protein L30 [67], eukaryotic initiation factor 4a3 [68], and nucleolin [69] have been predicted to act as facilitators or modulators of selenoprotein synthesis.

### 2.2. Selenoproteins

In total, 24 selenoproteins have been identified in mice, and the targeted deletion of some of these proteins demonstrated their key functions in disease pathophysiology [70]. About 25 different selenoprotein-coding genes have been identified in humans so far, and these include those encoding glutathione peroxidases (GPx1, 2, 3, 4, 6), thioredoxin reductases (TrxR1, 2, 3), iodothyronine deiodinases (DIO1, 2, 3), selenophosphate synthetase (SPSHS2), methionine-R-sulfoxide reductase 1 (MSRB1), and selenoprotein H, I, K, M, N, O, P, R, S, T, V, and W (SELENOH, SELENOI, SELENOK, SELENOM, SELENON, SELENOO, SELENOP, SELENOR, SELENOS, SELENOT, SELENOV, and SELENOW) [71,72,73]. These can be classified into subfamilies according to their cellular functions, which include those involved in antioxidation (GPx1, 2, 3, 4), redox regulation (TrxR1, 2, 3, MSRB1, SELENOH, M, W), thyroid hormone metabolism (DIO1, 2, 3), selenium transport and storage (SELENOP), selenophosphate synthesis (SEPHS2), calcium metabolism (SELENOK, T), myogenesis (SELENON), protein folding (SELENOF, I, S), and protein AMPylation (SELENOO). The functions of other selenoproteins, such as GPx6 and SELENOV, remain unclear [74,75,76]. GPx requires selenoproteins when it reduces lipid hydroperoxides to their corresponding alcohols and protects the organism from oxidative damage [77]. TrxR is a family of selenium-containing pyridine nucleotide-disulfide oxidoreductases involved in defense against oxidative damage via oxygen metabolism [78]. Most thyroid hormone deiodinases, which activate or deactivate various thyroid hormones and their metabolites, use selenoproteins as cofactors [79]. Hashimoto’s thyroiditis is induced when thyroid cells are destroyed by autoantibodies. Although the effect of dietary selenium on thyroid autoantibodies has been controversial, there is evidence that selenium intake reduces anti-thyroid peroxidase antibodies in patients with this disease [80]. The liver produces and dissolves SELENOP (also known as SePP), which acts as the primary transporter of Se to peripheral tissues. SELENOP concentration in the serum or plasma has been shown to be a valuable diagnostic parameter for selenium deficiency [81,82,83]. Most selenoproteins are beneficial for human diseases, indicating that the occurrence of a deficiency condition may be associated with the development or progression of pathological diseases [84]. However, the role of individual proteins must be integrated within a complex biochemical environment characterized by antagonistic, activator, and synergistic effects. The major characteristics of human selenoproteins, their roles in human health, and their correlation with diseases are summarized in Table 1.

### 2.3. Selenium Deficiency and Toxicity

Selenium is a microelement that rarely occurs in its elemental form in nature; instead, organic and inorganic mineral forms of selenium, such as selenide, selenate, and selenite, have been detected in soil [118,119,120]. Previous studies have shown that humans can absorb organic compounds better than inorganic compounds [121,122,123]. Inorganic forms of selenium, such as selenite, are used as dietary supplements [124]. Selenate is the most common inorganic form of selenium in soil; it dissolves in water and reaches rivers and oceans [125]. Organic forms of selenium occur mostly in plant and animal food as selenomethionine and selenocysteine [120,126]. Many whole grains, dairy products, meat, and seafood are good sources of selenium, although geographic variation in selenium content in soil may determine the level of selenium in food [127].

The margin between selenium deficiency and toxicity is narrow. A small amount of selenium is necessary for cell homeostasis because of its antioxidant activities, while supra-nutritional levels of selenium act as pro-oxidants in vivo [128]. Selenium poisoning, termed selenosis, can develop if selenium intake exceeds the tolerable limit of 400 µg/day [129]. Selenosis is characterized by garlicky breath, gastrointestinal disorders, hair loss, nail discoloration, fatigue, irritability, and neurotoxic effects [130]. Recent studies have suggested that excess selenium intake may increase the risk of type 2 diabetes [131]. Hydrogen selenide, a corrosive gas, is the most toxic selenium compound [132]. Interestingly, consumption of lower doses of selenium, such as 300 µg/day, may also exert negative effects on the body. Long-term consumption of 300 μg/day in the form of selenium-enriched yeast increased mortality in the cohort of the Danish Prevention of Cancer by Intervention with Selenium (PRECISE) trial. Hence, high daily selenium supplementation or intake should be avoided [133].

According to the World Health Organization (WHO) standards, the recommended daily intake of selenium for adults is 55 µg/day [51]. The optimal selenium concentration in plasma has been suggested to be 90–120 g/L, which is sufficient to saturate selenoproteins in blood [134]. People living in low selenium regions and those with compromised bowel absorption and undergoing chronic hemodialysis are at high risk of developing selenium deficiency [135]. Keshan disease, caused by a combination of dietary deficiency of selenium and Coxsackie virus infection, was first noted in people living in Northeast China, where soil contains low levels of selenium [136,137,138]. The affected patients suffer from congestive cardiomyopathy and pulmonary edema, often culminating in death [139,140]. Selenium deficiency has been suggested to be related to immune reactions against viruses [141,142]. People with Kashin-Beck disease (KBD) are distributed from Northeastern to Southwestern China. KBD is probably caused by the ingestion of fungal-contaminated grains [143] and is characterized by pain, limited motion in many joints of the body, and growth retardation in children, and selenium deficiency is considered a predisposing factor for the KBD [144]. Plasma selenium levels in children with KBD were lower than 27 ng/mL in the Tibet Autonomous Region, whereas appropriate selenium concentrations for optimal GPx activity are 80–100 ng/mL [33,145,146,147].

## 3. Roles of Selenoproteins in Bone Cells

Compared to boron, iron, zinc, and copper, selenium is a trace element present in bones that affects bone metabolism [1]. Bone has the second-highest proportion of body selenium (16%), followed by skeletal muscles (27.5%) [148]. Selenium deficiency is associated with low bone mass in male rats [149] and osteoarthropathy in KBD [150,151], as it adversely affects the biosynthesis of several antioxidant selenoproteins, which impairs bone metabolism [152].

### 3.1. Bone Remodeling

Healthy skeletal remodeling is maintained by an equilibrium between the activities of mesenchymal stem cell-derived osteoblasts and hematopoietic cell-derived osteoclasts on the bone surface [153]. The molecular triad of the receptor activator of the NF-κB ligand (RANKL) expressed by osteoblast lineage cells, osteoprotegerin (OPG) produced by osteoblastic or stromal cells, and the receptor activator of NF-κB (RANK) expressed by osteoclasts forms an important signaling link between osteoblasts and osteoclasts [154,155,156]. The RANK/RANKL pathway has been identified as the major signaling mechanism responsible for osteoclast differentiation. OPG is a non-signaling decoy receptor for RANKL that plays a critical regulatory role in osteoclast-mediated bone resorption by suppressing osteoclastogenesis and regulating mature osteoclasts [157,158]. Reactive oxygen species (ROSs) act as critical intracellular signaling mediators for RANKL-induced osteoclastic differentiation. Large amounts of ROSs are produced during bone remodeling, which are harmful for normal bone physiology, as they suppress osteoblastic differentiation and promote osteoclastic differentiation independent of NF-κB activation [159,160].

### 3.2. Selenoproteins in Bone Metabolism

The majority of selenoprotein-related genes and at least nine selenoprotein biosynthesis-related genes have been identified in osteoblasts or osteoclasts [161,162,163,164]. Figure 1 shows the mechanism underlying the metabolic effect of selenium on osteoblasts and osteoclasts.

TrxR1 is a selenoprotein that is upregulated by 1,25(OH)2D3 and is expressed early in the osteoblast differentiation signaling cascade, highlighting a part of the mechanism by which selenium promotes bone formation [45]. 1,25(OH)2D3 did not increase TrxR activity in human fetal osteoblast (hFOB) cells cultured in selenium-deficient media. The co-treatment of 1,25(OH)2D3 and selenium increased TrxR1 protein and activity, suggesting that 1,25(OH)2D3-mediated osteoblastic differentiation and/or maintenance of the differentiation program would be impaired if selenium was deficient in bone [165]. GPx1 is a critical antioxidant enzyme in osteoclasts that can inhibit osteoclastogenesis when activated [45]. Bone marrow matrix cells cultured at low selenium concentrations poorly expressed GPx and TrxR, and showed signs of chromosomal damage, which were restored by selenium supplementation; in addition, basal selenoprotein activity was recovered [162].

Additionally, evidence indicates that supplementing with supranutritional selenium can modulate specific pathological changes via various mechanisms, including GPx and TrxR suppressing NF-κB activation and its inflammatory reactions. [166,167,168,169]. Genetic studies have shown that a single nucleotide polymorphism (SNP) at codon 198 of GPx1 is associated with low bone mineral density and increase in the levels of bone turnover markers [170].

Adequate selenium consumption appears to be important for osteoclast and osteoblast cell proliferation and differentiation, mostly via ROS regulation. Studies have shown that extracellular signal-regulated kinases, ERK1/2, mediate the inhibitory effect of hydrogen peroxide on osteoblast differentiation [171,172,173]. Selenium treatment has been shown to protect bone marrow stromal cells from the hydrogen-peroxide-induced suppression of osteoblastic differentiation by inhibiting oxidative stress and ERK activation [174,175,176]. In contrast, ROS levels increased when selenoprotein expression decreased, which was related to pathologically exacerbated signaling and enhanced osteoclast activity [163,177].

Recently, Kim et al. demonstrated that SELENOW contributes to osteoclastogenesis, but is downregulated via RANKL/RANK/tumor-necrosis-factor-receptor-associated factor 6/p38 signaling. They suggested that this RANK-dependent inhibition of SELENOW blocks overactive osteoclasts to prevent osteoporosis [178]. SELENOP is a major selenium transporter in plasma that is produced and released by hepatocytes and resorbed by bone cells via apolipoprotein receptor 2(ApoER2) [179,180,181]. In SELENOP-mediated tissue-specific selenium absorption, transcription of mRNA and LDL-receptor-related protein 8 (LRP8) protein expression has been proceeding. A crucial component of the intracellular selenium transportation system, the LRP8/ApoER2 complex is increased in chronic selenium deficiency, suggesting this process as important feedback to avoid chronically low SELENOP levels [46,161]. Interestingly, bones, brain, and testis are resistant to limited selenium supply, ensuring that the most essential selenium-dependent processes continue in specific tissues even under selenium deficiency due to increases in the expression of ApoER 2 [180,181,182].

## 4. Selenium and Bone: Animal Model Studies

Few studies have investigated the effects of selenium on bone metabolism in animal models [183,184,185]. Table 2 presents the results of animal studies regarding intervention with selenium supplementation.

Decades ago, Sasaki et al. discovered that a low-selenium diet of 3–11 months reduced bone volume, bone mineral density, and femur ash weight in rats [186]. Turan et al. [187] fed young rabbits of both sexes with a selenium- and vitamin-E-adequate diet (control group), a selenium- and vitamin-E-deficient diet, or excessive selenium for 12 weeks. They evaluated three-point bending, representing the risk of bone fracture [188]. In this study, plasma red blood cell GPx activity increased or decreased relative to plasma selenium levels in the group fed a selenium- and vitamin-E-deficient diet and the group fed a selenium-excess diet. Although selenium-deficient diets did not affect body weight or growth patterns, the biomechanical strength of the bones decreased significantly in the groups fed the selenium- and vitamin-E-deficient diet or the selenium-excess diet. Notably, bone histopathology observed in the selenium- and vitamin-E-deficient group was indicative of osteomalacia [187]. Osteomalacia is a disorder of decreased bone mineralization with a histological feature that is different from osteoporosis, which shows normal bone mineralization but an ongoing loss of bone matrix that leads to decreased bone mass [189]. Later, Turan et al. [190] developed a heparin-induced osteoporosis model in adult female New Zealand white rabbits. They were divided into three experimental and control groups, each on the basis of supplementation with deionized water, a combination of vitamin C and E, and sodium selenite plus a combination of vitamin C and E. The long bone tissue of rabbits from the group fed selenium plus combined vitamin supplementation showed almost the same structure as that of normal rabbits. [190]. Similarly, Cao et al. administered a four-month selenium-deficient or -adequate diet to adult male C57BL/6J mice and reported that bone microarchitecture was impaired in the selenium-deficient diet group, accompanied by increases in the levels of bone absorptive markers and decreases in GPx activity in the blood [177]. Yao et al. examined the effects of supplemented selenium combined with iodine, which was designed for a diet for the KBD endemic area, on the histology of bone and growth plate cartilage in 96 Wistar rats of both sexes [191]. Supplementation with selenium and iodine decreased necrosis of the chondrocytes in the growth plate and promoted trabecular bone formation in rats. Additionally, they observed increases in bone volume/tissue volume ratio, trabecular thickness, and trabecular number and decreases in trabecular separation [191].

The effects of a selenium diet on bone metabolism appear to be clear in animal studies conducted over generations [149,192,193]. Moreno-Reyes et al. provided a selenium-depleted diet to growing male rats for two generations [149] and observed growth retardation and low bone mineral density in the second-generation rats fed a selenium-depleted diet compared with the first-generation rats. The selenium-depleted diet was associated with reductions in the levels of pituitary growth hormone and insulin-like growth factor I [149].

Ren et al. designed an animal study where a selenium- and/or iodine-deficient diet was supplied for two generations and reported that the combination of selenium and iodine deficiency impaired bone and cartilage growth [192]. Additionally, Min et al. showed that selenium deficiency in rats fed a low-selenium diet over two generations can lead to epiphyseal plate thickness shortening, pathological changes, and decreased antioxidant activity [193].

**Table 2 molecules-27-00392-t002:** Effects of selenium on bone health in in vivo studies.

Author, Year [Ref]	Study Design	Animal Model	Intervention (Dose, Formula, Duration)	Result
Turan et al. 1997 [187]	Assigned to 4 groups: Se- and vitamin-E-adequate/ Se- or vitamin-E-deficient Se-excess diet.	New Zealand white rabbits	Normal Se diet, 0.5 mg/kg; excess Se diet, 10 mg/kg. Sodium selenite. 12 weeks.	GPx activity ↑ or ↓: plasma selenium level ↑ or ↓. Light microscopic investigations of the bone tissues showed osteomalacia. Biomechanical strength of the bones ↓ in both Se-deficient and Se-excess diets.
Moreno-R et al. 2001 [149]	Assigned to Se-deficient diet for two generations.	Wistar rats	Se-deficient diet, 0.005 mg/kg; Se-adequate diet, 0.19 mg/kg. 72 days.	Weight and tail length ↓. Plasma calcium ↓ and urinary calcium concentration ↑. PTH and 1,25(OH)2D3 2-fold ↑. Plasma osteocalcin and urinary deoxypyridoline ↓. BMD of the femur and tibia ↓.
Turan et al. 2003 [190]	1000 IU/kg/day heparin for 4 weeks. Osteoporosis model.	New Zealand white rabbits	Se diet, 0.05 mg/kg/day. Sodium selenite. 30 days.	Combination of vitamins E and C with selenium prevented structural alterations in the long bones in an osteoporosis model.
Ren et al. 2007 [192]	Assigned to 4 groups: Se- and iodine-sufficient/-deficient diets for two generations.	Sprague-Dawley rats	Se-deficient diet, <0.02 μg/g; Se-adequate diet, 0.1–0.3 μg/g. 3 months.	Tibial length, the thickness of the growth plate cartilage ↓ in Se- and iodine-deficient rats. PTHrP expression ↑. ColX expression ↓ in Se-deficient rats in both generations, independently of iodine deficiency.
Cao et al. 2012 [177]	Assigned to 3 groups: purified, Se-deficient, and Se-adequate diet.	C57BL/6J mice	Se-deficient diet, 0.9 μg Se/kg; Se-adequate diet, 100 μg Se/kg. SeMet or SeBean. 4 months.	In selenium-deficient group: GPx1 activity and GPx1 mRNA in liver ↓. Femoral trabecular bone volume/total volume ↓, and trabecular separation ↑. Bone structural parameters ↔. Serum CRP, TRAP, and PTH ↑.
Yao et al. 2012 [191]	Assigned to 4 groups: control, KBD diet, Se diet, and Se plus iodine diet.	Wistar rats	KBD Se diet, 0.031 mg/kg; normal Se diet, 0.14 mg/kg. 12 weeks.	Selenium supplement group and selenium plus iodine group: bone volume/tissue volume ratio (BV/TV), trabecular thickness, trabecular number ↑, trabecular separation ↓. KBD diet group: Tibial growth plate chondrocyte necrosis ↑, total serum protein and albumin ↓.
Min et al. 2015 [193]	Assigned to 2 groups: Se-sufficient/-deficient diets for two generations.	Dark Agouti rats	Se-sufficient diet, 0.288 μg/g; Se-deficient diet, 0.018 μg/g. 2 months.	In two generations of rats, gene expressions of COL II, GPx1, and GPx4 ↓ in Se-sufficient rats. Epiphyseal plate lesion ↓, cartilage type II collagen production ↓, and GPx1 activity ↓.

Abbreviations: SeMet: Selenomethionine; SeBean: seleno pinto beans; Se: selenium; TRAP: tartrate-resistant acid phosphatase; KBD: Kashin-Beck disease; 1,25(OH)2D3: 1,25-dihydroxyvitamin D3; PTH: parathyroid hormone; Gpx: glutathione peroxidase; BMD: bone mineral density; PTHrP: parathyroid hormone-related peptide; COL II: type II collagen; ColX: type X collagen; ↑ indicates increase or upregulation; ↓ indicates decrease or downregulation; ↔ indicates no change.

## 5. Clinical Studies on the Effects of Selenium on Bone Health

As observed in preclinical studies, selenium deficiency may potentially cause bone damage. Despite efforts to prove the association between selenium and bone health in humans, clinical studies have yielded controversial outcomes. An extensive literature search was done on correlations between bone health, selenium blood level, and selenium dietary intakes and on the effectiveness of selenium supplementation in humans. For the present section, we reviewed a total of 21 studies: 6 case–control studies, 10 cross-sectional studies, 2 cohort studies, 1 longitudinal study, and 2 randomized, double-blind, placebo-controlled studies. Table 3 and Table 4 compare the results of previous observational, clinical studies, and they were grouped according to the ethnicity of the population and are listed in order. Most of these observational studies assessed selenium status by measuring blood selenium levels alone or paired with selenoprotein P levels, or by using a questionnaire on selenium intake. The outcomes of bone health were considered by BMD or fracture history.

### 5.1. Association of Selenium Intake Levels with Bone Health: Observational Studies

Several case–control studies have been performed to determine the differences in selenium levels between patients with osteoporosis and healthy individuals. In a study which is the first of its kind, the differences in the concentrations of elements, including magnesium, zinc, copper, manganese, and selenium, were determined in 77 postmenopausal women with osteoporosis and 61 healthy controls. However, the plasma concentrations of selenium and other elements did not vary significantly between the patients and controls [194]. Liu et al. investigated the serum concentrations of macro and trace elements such as zinc, iron, copper, and selenium in 290 postmenopausal women in the Chinese urban area. Comparison of bone mineral density of healthy individuals with those with osteoporosis and osteopenia did not reveal any significant difference in the levels of these elements [198]. Wang et al. grouped 91 elderly Chinese men with osteoporosis and showed that plasma levels of Mn, Zn, Cu, Se, and Pb were not related to bone mineral density [199]. Some studies have documented that the levels of triglycerides and low-density lipoprotein cholesterol correlated negatively with selenium content in postmenopausal women [209,210]. There were no differences in the levels of trace elements such as selenium and lipids in 107 women who were divided into postmenopausal osteopenia, osteoporosis, and health disorder groups, and there was no direct correlation between selenium and bone mineral density [47].

Other population-based cohort studies have used large numbers of participants. A prospective study [48] that characterized the association between selenium status and bone mass among 1144 healthy euthyroid postmenopausal women demonstrated that plasma selenium levels or SELENOP concentrations correlated with hip and lumbar spine bone mineral density and bone turnover markers. Higher selenium status was related to low bone turnover markers and high bone mineral density, but was not related to a risk of vertebral fracture [48]. Beukhof et al. determined selenium content by measuring both plasma selenium and SELENOP levels in a cohort study of 387 healthy elderly men in the Netherlands. Selenium and SELENOP levels were positively associated with total and femoral trochanter bone mineral density in subjects without selenium deficiency [195]. Al-E-Ahmad et al. showed an association between selenium level and lumbar spine and femoral neck bone mineral density among elderly individuals living in Iran. As selenium possesses antioxidant activity, they investigated the correlation between SNPs in ROS-related genes [211,212,213] and selenium levels and revealed that selenium reduced ROS activity. In addition, selenium levels correlated positively with bone mineral density [197].

Recent studies have shown a positive correlation between selenium levels and bone mineral density. In a study on 1167 healthy Korean adults, Park et al. found differences in hair selenium levels between the low and normal bone mineral density groups. Participants with low bone mineral density had significantly lower hair selenium levels adjusted for osteoporosis-related risk factors compared to a healthy control [200]. In the population-based Hortega study [196] involving 1365 participants, a non-linear dose–response relationship of selenium was observed with reductions in bone mineral density, showing an inverse association below 105 μg/L, which became increasingly positive above 105 μg/L. While a U-shaped dose–response curve of selenium was observed with the incidence of osteoporosis-related bone fractures in a prospective analysis, the positive association above 105 μg/L was markedly stronger than that observed in the cross-sectional analysis [196]. In a study that did not directly evaluate selenium levels, Mendelian randomization analysis was performed on summary data from large Genome-wide Association Study (GWAS) datasets, and serum selenium levels were found to correlate positively with the heel bone mineral density. A nominally significant relationship was observed between serum selenium levels and forearm bone mineral density. However, selenium significantly affected only bone mineral density at specific skeletal sites such as heel [214].

Other studies have investigated selenium intake instead of measuring selenium levels in the blood. Wolf et al. measured bone mineral density in 11,068 postmenopausal women aged 50–79 years enrolled in the Women’s Health Initiative Observational Study and Clinical Trial at three clinics in the USA, where the level of selenium was estimated using a self-reported food-frequency questionnaire. However, dietary selenium intake and total bone mineral density were not associated [49]. Two more studies showed a lack of correlation between selenium intake and bone mineral density [202,205]. In contrast, several studies have shown a positive correlation with bone health. Zhang et al. constructed a case–control study and showed that selenium intake was inversely associated with a reduced risk of osteoporotic hip fracture in 2564 elderly smokers (OR = 0.27, 95% CI = 0.12–0.58) [201]. An earlier case–control study also showed that a higher dietary intake of selenium was associated with a lower risk of hip fracture (OR = 0.43, 95% CI = 0.26–0.70) [206]. Two Spanish studies of postmenopausal and non-menopausal women showed that a selenium diet was associated with bone mineral density at specific sites, including the calcaneus or phalanges [203,204].

In a cross-sectional study of 6267 Chinese people, the prevalence of osteoporosis was high (OR = 0.72, 95% CI = 0.55–0.94) in both men and women with lower dietary selenium intake levels. Although several confounding factors were considered in this study, unfortunately, detailed information on hormonal status is lacking [207]. Zhang et al. used data from 17,150 participants from the China Health and Nutrition Survey, in which the self-reported history of fractures showed a non-linear association with selenium intake [208]. Recently, Wu et al. analyzed the data of 2983 adults in the National Health and Nutrition Examination Survey (NHANES) 2013–2014 [50] and observed that a higher selenium level, measured in plasma and diet, correlated with a lower bone mineral density and a decreased FRAX score, which represents the 10-year fracture risk [50].

Although there are studies showing non-significant results between selenium and bone health [47,49,194,198,199,202,205], a positive correlation was observed between selenium and bone health, showing a positive relationship between selenium and bone mineral density [48,50,195,196,197,200,203,204,207]and a reduction in the risk of osteoporotic fracture [50,196,201,206,208]. The limitations of these observational studies include the fact that not all, but some, were conducted with sample sizes too small to achieve statistically valid results. The majority of studies found no evidence of a selenium deficiency, as demonstrated by the Recommended Daily Allowance (RDA).

### 5.2. Randomized Clinical Trial of Selenium Intake: Intervention Studies

Based on the positive results obtained in the above observational studies, two randomized clinical trials were performed. The characteristics of interventional studies are shown in Table 5. The effect of selenium level on bone health was tested for the first time in a randomized controlled trial by Walsh et al., where they completed a 6-month randomized, double-blind, placebo-controlled trial involving postmenopausal women aged >55 years with osteopenia or osteoporosis in the UK. An equal number of participants were randomly assigned to a placebo or to groups administered 50 μg or 200 μg of oral sodium selenite per day. Among the 120 participants recruited, 115 (96%) completed the follow-up. The primary endpoint was a biochemical marker of bone resorption (urinary *N*-telopeptide, uNTx) at 26 weeks, the levels of which did not differ significantly between the treatment groups. Secondary assessments of other markers of bone turnover, i.e., bone mineral density, muscular performance, and levels of glutathione peroxidase, highly sensitive C-reactive protein, and interleukin-6, revealed that there were no significant changes after selenium supplementation [215]. The authors of that study mentioned that the dietary selenium intake at baseline was higher than that expected in the UK population. Second, the study population included only postmenopausal women, suggesting that further studies investigating elderly people with selenium deficiency are required [215].

A population with relatively low selenium level was investigated in a single-center, randomized, double-blinded, placebo-controlled, multi-arm, parallel clinical trial (PRECISE) in Denmark [216]. The 491 volunteers, aged 60–74 years, were randomly and equally assigned to treatment groups administered 100, 200, or 300 µg selenium per day via selenium-enriched yeast for 5 years (from randomization in 1998–1999) and were followed up for further 10 years. The plasma selenium level was within the optimal level during the follow-up period. The levels of bone turnover markers, measured at 6 months, showed a significant linear decrease in the level of procollagen type 1 *N*-terminal propeptide (P1NP), although this effect was not apparent at 5 years; no significant effect of selenium supplementation on any other bone formation markers was observed [216]. As P1NP is a marker of osteoblast function [217], the decrease in PINP level indicated a reduction in bone formation. Therefore, the effect of selenium on bone turnover and bone health remains to be determined on the basis of the results of these randomized clinical trials.

## 6. Novel Approach of Selenium Supplement

The contradictions in the results of clinical studies can be because of the differences in the levels of dietary selenium and selenium supplements [218]. Therefore, current studies have focused on developing various selenium-based therapeutic strategies instead of traditional selenium intake trials. Here, we have focused on various novel therapeutic strategies involving selenoproteins.

### 6.1. Selenium Nanoparticles

The amount of bioavailable selenium in the diet largely differs with the food and soil fodder to which humans and animals are exposed [219,220]. Moreover, in the environment, selenium is present in various oxidation states (2, 0, 4+, and 6+) that are cytotoxic. Owing to its high toxicity (Se^4+^), selenite is reduced to elemental selenium (Se^0^) by biogeochemical cycles [221,222,223]. Therefore, a high dose of selenium is harmful for humans and has a narrow safety-to-toxicity margin [224,225]. Selenium nanoparticles are potential candidates with low cytotoxicity [226,227] that can be used to circumvent the various problems associated with inorganic and organic selenium compounds, such as ROS generation and low redox activity [228,229,230,231].

#### 6.1.1. Synthesis of Selenium Nanoparticles

Selenium nanoparticles can be synthesized chemically [232], physically [233], or even biologically using microorganisms or plant extracts, a process known as green synthesis [234,235]. Gao et al. [236] showed that hollow, chemically synthesized, spherical selenium nanoparticles possess antioxidant properties that can reduce selenium toxicity. Selenium nanoparticles are less toxic than inorganic and organic selenium. The lower hepatotoxicity and nephrotoxicity of these nanoparticles have been demonstrated in in vivo studies [226,237,238]. However, these chemically synthesized nanoparticles are not suitable for biomedical use, limiting their application. As harsh chemicals are used in chemical synthesis, even small amounts of these compounds in humans can cause systemic toxicity. Therefore, green synthesis is gaining popularity in the biomedical field [239,240,241].

Furthermore, oral supplementation with nanoparticles is considered to be the most appropriate and cost-effective method. However, absorption barriers in the digestive tract may render the absorption of nanoparticles difficult. For example, to metabolize nanoparticles, the mucus covering the intestinal mucosa should be overcome [242]. Thus, numerous compounds have been used to stabilize selenium nanoparticles, including amino acids, bovine serum albumin, polysaccharides such as chitosan [243,244,245], gum arabic [243,246], and polymers such as poly(lactic-co-glycolic acid) [247].

#### 6.1.2. Effect of Selenium Nanoparticles on Bone Health

Selenium nanoparticles have been shown to possess therapeutic potential against various cancer cells, microbial pathogens, and viruses and protect from hepatic diseases [218,248,249,250], diabetes [251,252,253], and neurological diseases [254,255,256]. However, few studies have reported the therapeutic effects of selenium nanoparticles on bone-related conditions.

Experiments have been performed with green selenium nanoparticles. Polysaccharide-protein complex-coated selenium nanoparticles (PTR-SeNPs) synthesized using the mushroom polysaccharide isolated from *Pleurotus tuberregium* and selenium nanoparticles (Cs4-SeNPs) synthesized using the mushroom polysaccharide isolated from *Cordyceps sinensis* inhibited osteoclastogenesis while promoting osteoblastogenesis, thereby stimulating bone formation via preosteoblast differentiation [257,258]. The PTR-SeNP system has been rationally designed and identified as a potent bone-formation therapeutic that can antagonize osteoporosis. The nanosystem is incorporated by osteoblasts and significantly enhances bone formation in vitro and in vivo [259]. Fatima et al. [260] found that modest concentrations of selenium nanoparticles can improve human mesenchymal stem cell survival and osteogenic potential by reducing oxidative stress. Selenium nanoparticles may promote osteogenesis by activating the JNK/FOXO3 pathway. Another study also demonstrated increases in the osteogenic maturation of MC3T3-E1 cells and the downregulation of ROSs by hydrogen peroxide [261].

Many clinical studies have investigated the effects of cancer chemotherapy on osteoporosis. Anastrozole is used to treat breast cancer; however, bone toxicity limits its clinical use. In vitro studies on human osteoblasts have shown that selenium nanoparticles reduce cell death. At the examined doses, selenium nanoparticles significantly prevented decreases in bone density and increases in the levels of biochemical markers of bone resorption, eventually preventing anastrozole-induced osteoporosis [262]. Therefore, selenium nanoparticle co-supplementation might prevent bone loss by enhancing osteogenesis and may be an effective candidate for treating osteoporosis.

### 6.2. Implanted Selenium Therapeutics

Numerous metal ions, including selenium, possess osteoinductive properties. Hence, the effects of coating or implanting selenium and selenium nanoparticles on bone structures and tumors [263], or other orthopedic medical devices [264], have been investigated. The integration of selenium into biomaterials is an effective therapy for promoting bone remodeling [265,266].

Previous studies have shown that the local use of selenium correlates with bone formation and reduces the risk of osteoporosis due to its antioxidant property [175,267]. The effect of calcium phosphate ceramics on bone healing [268] and the strong affinity between selenium and calcium phosphate have been shown [266]. Li et al. [269] conducted a study with selenium-modified calcium phosphate cement (Se-CPC), CPC with 3% selenium nanoparticles, and scaffolds. These were implanted into the femoral epiphysis of an ovariectomized rat with osteoporosis and incubated for 12 weeks. The results of a reverse transcription-quantitative polymerase chain reaction revealed that the Se-CPC group had higher OPG and lower RANKL levels. This stimulated bone regeneration and bone formation by activating the OPG/RANKL pathway and inhibiting local oxidative stress [269]. Additional experiments should be performed in the future to expand the availability of localized selenium nanoparticles.

## 7. Conclusions and Future Prospects

The effects of selenium on human health are numerous and complex, and there are significant information gaps that have to be filled. In the present review, we highlighted the importance of selenium in maintaining bone health, especially in osteoporosis, which is one of the major diseases affecting bone health.

Osteoporosis contributes significantly to a social management burden due to the high mortality, morbidity, and treatment costs associated with the disease [270]. By 2020, the number of people suffering from osteoporosis worldwide exceeded 200 million, which will increase rapidly in successive years. The number of annual hip fractures will surpass 3 million, and the cost of treatment is estimated to be 25 billion USD by 2025 [13,271].

New drugs are available for the treatment and prevention of osteoporosis. As mentioned earlier, the daily intake of calcium- and vitamin-D-rich food is part of the traditional lifestyle modification therapy [272]. Only subtle adverse events have occurred with anti-osteoporotic drugs including romosozumab, calcium supplements, bisphosphonates, and selective estrogen receptor modulators, which are currently safely prescribed and widely used [273,274,275]. In addition to anti-osteoporotic drugs, natural products, such as trace elemental particles that are safe for human health, can be used in conjunctive therapy. Selenium may be used as an adjuvant for the treatment of osteoporosis, as some studies have demonstrated that selenium exerts cardiovascular protective and anti-osteoporotic effects [276,277,278,279,280].

Most observational studies have reported that the level of selenium correlates positively with bone health, as it increases bone mineral density or reduces the risk of osteoporotic fracture, while some studies have shown a lack of association between selenium status and bone health. However, there is a discrepancy between the results of randomized clinical trials and those of observational studies. The baseline selenium level or the formula of selenium supplement might be a key factor influencing the effect of selenium on bone health. Further research on the different functions and effects of selenium biomarkers and on more effective and safe methods of selenium supplementation are required. Therefore, the novel approach of synthesizing selenium supplements, such as selenium nanoparticles, is being investigated.

Nanobiotechnology has been a hot topic of research ever since its emergence in the biomedical domain. Green nanoparticles are produced using natural microbes, which can be optimal for human use. However, reports on green-synthesized selenium nanoparticles are limited. Further studies are required to understand the relationship between selenium and selenium nanoparticles, as well as the biological pathways underlying the differences in the therapeutic efficacies of selenium and selenium nanoparticles. In the future, the use of selenium nanoparticles should be expanded to treat bone health and other medical conditions. Finally, the effect of selenium on bone health is still an open area of investigation, and more studies on the role of selenium nanoparticles are required.

## Figures and Tables

**Figure 1 molecules-27-00392-f001:**
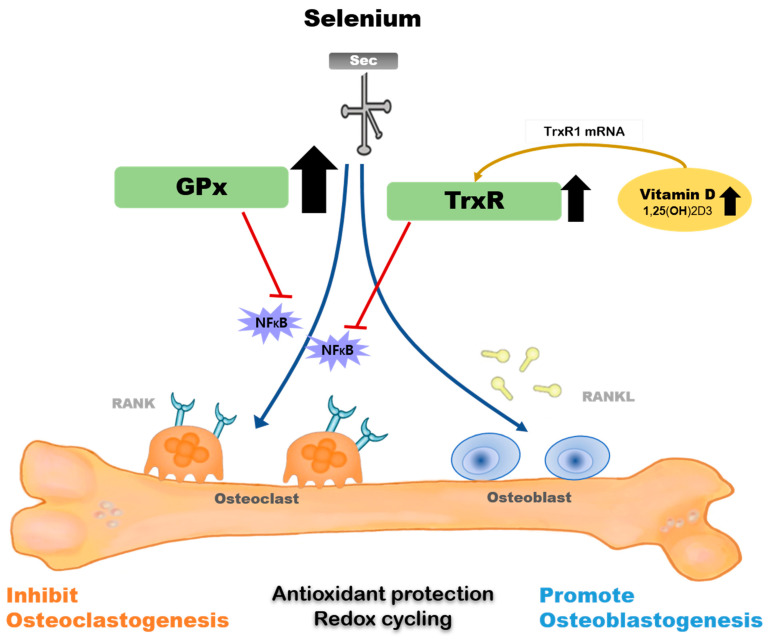
Mechanism of selenium-related bone metabolism. Selenium supplement encourages the activity of GPx and TrxR, and GPx activity is more activated in those with a selenium-deficient status. GPx and TrxR suppress NFκB activation at supranutritional selenium levels and regulate osteoclastogenesis and osteoblastogenesis. Co-treatment of 1α,25(OH)2D3 and selenium synergistically elevates TrxR1 protein and activity. GPxs: glutathione peroxidases; TrxRs: thioredoxin reductases; RANK: receptor activator of NFκB; RANKL: receptor activator of NFκB ligand.

**Table 1 molecules-27-00392-t001:** Selenoprotein classified by cellular function and effects on health [29,61,84,85,86,87,88,89,90].

Selenoprotein	Subcellular Location	Function	Effect on Health and Disease	Ref.
Glutathione Peroxidases (GPxs)				
GPx 1	Cytoplasm Mitochondrial membrane	Antioxidant. Cellular H_2_O_2_ ↓and lipid peroxide ↓. Cardioprotective and anti-angiogenic.	Cardiovascular: CVD, hypertension, peripheral vascular disease, ICH. Cancer: lung, prostate, bladder, primary liver Keshan disease, Kashin–Beck disease.	[91,92,93,94,95,96]
GPx2	Cytoplasm	Antioxidant. Peroxide ↓ in the gut. Carcinogenesis.		
GPx3	Secreted abundantly in plasma	Antioxidant. Peroxide ↓ in blood. Tumor suppressor.	Ischemic stroke, differentiated thyroid cancer.	[97,98]
GPx4	Cytoplasm Mitochondria Nuclei	Antioxidant. Phospholipid peroxide ↓. Regulation of ferroptosis and phospholipid, H_2_O_2_ levels. Involved in spermiogenesis.	Adenomatous polyps, male infertility. Cancer: colorectal cancer, breast cancer.	[99,100]
GPx6	Secreted protein	Cellular H_2_O_2_ ↓ in the olfactory epithelium.	-	
Iodothyronine Deiodinases (DIOs)				
DIO1	Plasma membrane	Regulation of systemic circulating thyroid hormone levels.	Muscle strength, lean body mass. Cardiovascular diseases.	[101]
DIO2	ER membrane	Regulation of local muscular thyroid hormone levels.	Type-2 diabetes and insulin resistance. Osteoporosis and osteoarthritis. Mental retardation.	[102,103,104,105]
DIO3	Plasma membrane	Inactivates thyroid hormone.	Osteoporosis and osteoarthritis.	[106]
Thioredoxin Reductases (TrxRs)				
TrxR1	Cytoplasm Nucleoplasm	Antioxidant. Regenerates reduced thioredoxin. Embryonic development.	Advanced colorectal adenoma.	[107]
TrxR2	Mitochondrial membrane	Antioxidant. Catalyzes a variety of reactions, specific for Trx and GPx. Embryonic development.	Gastric cancer.	[108]
TrxR3	Cytoplasm Nucleoplasm	Antioxidant. Oxidized ↓ of Trx and GPx2. Involved in spermiogenesis.	-	
Selenoprotein P (SEPP)	Secreted protein	Selenium transport to peripheral tissues.	Prostate cancer, colorectal adenoma, colorectal cancer. Neurological disorders, type 2 diabetes.	[100,107,109,110]
Selenoprotein M (SELENOM)	ER membrane	Protein folding in ER. Involved in redox regulation.	Neurological disorders.	
Selenoprotein N (SELENON)	ER membrane	Regulation of muscle development.	Muscle disorders.	[111,112]
Selenoprotein O (SELENOO)	Mitochondrial membrane	Possibly involved in redox regulation.	-	
Selenoprotein S (SELENOS)	ER membrane	Involved in ER-associated degradation and immune response. Involved in protein folding.	Cardiovascular disease. Cancer: gastric, colorectal, and rectal cancer. Inflammatory response. Pre-eclampsia.	[100,113,114,115]
Selenoprotein T (SELENOT)	ER membrane Golgi-complex	Involved in redox regulation and cell anchorage.	-	
Selenoprotein V (SELENOV)	Unclear	Unknown.	-	
Selenophosphate synthetase 2 (SEPHS2)	Nucleoplasm	Synthesis of selenophosphate.	-	
MSRB1 Selenoprotein R (SELENOR)	Cytoplasm Nucleoplasm	Antioxidant. Regulation of actin polymerization. Involved in redox regulation.	Cardiovascular diseases.	
Selenoprotein W (SELENOW)	ER membrane	Antioxidant. Involved in muscle growth and differentiation. Involved in redox regulation.	Muscle disorders, neurological disorders.	
Selenoprotein H (SELENOH)	Nucleoplasm and nucleoli	Antioxidant. Involved in redox regulation.	Neurological disorders.	
Selenoprotein I (SELENOI)	Plasma membrane	Involved in phospholipid biosynthesis. Involved in protein folding.	-	
Selenoprotein F (SELENOF)	ER membrane	Involved in protein folding. Possibly implicated in cancer etiology.	Cancer: Prostate cancer, lung cancer, and rectal cancer.	[113,116,117]
Selenoprotein K (SELENOK)	ER membrane	Modulates calcium metabolism. Involved in ER-associated degradation and immune response.	-	

Abbreviations: ER: endoplasmic reticulum; MSRB1: methionine sulfoxide reductase B1; H_2_O_2_: hydrogen peroxide; BAT: brown adipose tissue; ICH: intracerebral hemorrhage; ↓: reduced or decreased.

**Table 3 molecules-27-00392-t003:** Studies on blood selenium level and bone health.

Author, Year [Ref]	Setting/ Study Design	Country	Number of Subjects	Case/Control Mean Serum Selenium Status		Outcomes Considered	Results	+/-
Odabasi et al. 2008 [194]	Case–control study	Turkey	138 postmenopausal women	Osteoporosis (*n* = 77) Control (*n* = 61)	Se (ng/mL): 76.98 Se (ng/mL): 78.98	Trace element. With BMD.	No significant difference between the osteoporosis and control group.	-
Arikan et al. 2011 [47]	Case–control study	Turkey	107 postmenopausal women	Osteoporosis (*n* = 35) Osteopenia (*n* = 37) Healthy (*n* = 35)	Se (μg/L): 66.16 ± 12.1 Se (μg/L): 66.89 ± 15.5 Se (μg/L): 67.12 ± 11.6	Trace element. Lipid level. With BMD.	No significant differences between the osteoporosis, osteopenia, and healthy groups.	-
Hoeg et al. 2012 [48]	Population-based cohort study	Europe (OPUS study)	1144 postmenopausal women	Mean Se: 94.3 (μg/L) Mean SePP: 3.2 (mg/L)		Se, SePP, TFT, bone turnover markers (OC, PINP, uNTX to Cr). With BMD. Vertebral, hip, nonvertebral fracture.	Se and SePP statuses were inversely related to bone turnover markers. Positively correlated with BMD. Se and SePP statuses were not related to both non-vertebral and vertebral fracture.	+
Beukhof et al. 2016 [195]	Population-based cross-sectional study	The Netherlands	387 elderly men	Mean Se (μg/L): 91.9 Mean SePP (mg/L): 3.4 0.5% of subjects are Se-deficient		Se, SePP. TFT. With total and femoral BMD.	Se and SePP statuses were positively associated with total BMD and femoral trochanter BMD.	+
Marta et al. 2021 [196]	Cross-sectional, population-based cohort study	Spain (Hortega Study)	1365 >20 yr adults	Low BMD High BMD	Se level: 82.8 μg/L Se level: 85.7 μg/L	Se, As, and Cd. With calcaneus BMD and osteoporosis-related bone fractures (in >50 yr older subjects).	Calcaneus BMD had non-linear dose–response; inverse below 105 μg/L and positive above 105 μg/L. Osteoporosis-related bone fractures show U-shape dose–response; positive above 100 μg/L, HR 2.25 (1.13–4.49).	+
Al-E-Ahmad et al. 2018 [197]	Case–control study	Iran	180 elderly adults	Osteoporosis (*n* = 90) Healthy (*n* = 90)	Se (μg/L): 57.58 ± 25.5 Se (μg/L): 81.09 ± 25.6	ALOX12 SNPs. Se level. With BMD.	Se level was different among groups Positively correlation between serum Se and BMD.	+
Liu et al. 2009 [198]	Cross-sectional study	China	290 postmenopausal women	Osteoporosis (*n* = 123) Osteopenia (*n* = 127) Healthy (*n* = 31)	Se (mg/L): 0.067 ± 0.02 Se (mg/L): 0.069 ± 0.02 Se (mg/L): 0.065 ± 0.01	Serum macro-element and trace element. With BMD.	No significant differences between the osteoporosis, osteopenia, and healthy groups. No correlation between BMD and selenium.	-
Wang et al. 2015 [199]	Case–control study	China	91 elderly men	Osteoporosis (*n* = 30) Osteopenia (*n* = 31) Healthy (*n* = 30)	Se (ppb): 125.53 ± 22.8 Se (ppb): 144.88 ± 26.8 Se (ppb): 133.97 ± 29.0	Trace element. With BMD.	No significant differences between the osteoporosis, osteopenia, and healthy groups. No correlation between BMD and selenium.	-
Park et al. 2020 [200]	Cross-sectional study	Korea	1167 adults	Low BMD group Normal group	Se level: 0.05 μg/g Se level: 0.06 μg/g	Hair Se. Quartile level. With BMD.	Lower Se levels were associated with low BMD.	+

Abbreviations: BMD: bone mineral density; DXA: dual energy X-ray absorptiometry; Se: selenium; SePP: selenoprotein P; TFT: thyroid hormone test; PINP: procollagentype I *N*-terminal propeptide; OC: osteocalcin; uNTX to Cr: urinary resorption marker *N*-terminal telopeptide of type I collagen to creatine; SNPs: single nucleotide polymorphisms; OPUS study: Osteoporosis and Ultrasound Study; + means positive correlation with bone health; - means no significant correlation.

**Table 4 molecules-27-00392-t004:** Studies on selenium intake and bone health.

Author, Year [Ref]	Study Design	Country	Number of Subjects/Age	Diet Assessment	Outcome Considered	Selenium Intake/Selenium Status	Results	+/-
Wolf et al. 2005 [49]	Population-based cross-sectional study	USA, 3 clinical sites	11,068 postmenopausal women 50–79 years	Semi-quantitative FFQ	Intakes of antioxidants with total BMD	Antioxidant diet group Se: 85.9 ± 38.6 μg/d. Total group Se: 94.1 ± 43.2 μg/d.	Selenium had no association with BMD after multiple adjustments.	-
Zhang et al. 2006 [201]	Population-based case–control study	USA, Utah	1215 hip fracture 1349 controls ≥ 50 years	137-item FFQ	Intakes of antioxidants with hip fracture modified by smoking	Quintile of antioxidant intake of Se: 58, 79, 99, 121, 162 μg/d.	Selenium was associated with reduced risk of osteoporotic hip fracture. Selenium hip fracture OR 0.27 (0.12- 0.58) in ever smokers.	+
Wu et al. 2021 [50]	Cross-sectional, population-based cohort study	USA	2983 Adults ≥ 40 years	2-day food records FFQ	Whole blood and serum Se with FN, LS BMD, and FRAX score	Mean dietary selenium intake: 101.5 μg/day. Mean whole blood selenium: 196.7 μg/L. Mean serum selenium: 131.1 μg/L.	Increased Se status is correlated with an increased FN BMD, decreased FRAX scores, and reduced incidence of previous bone fracture history.	+
Ilich et al. 2008 [202]	Cross-sectional study	Croatia	120 postmenopausal women	3-day food records FFQ	Hip and LS BMD	Control group: 104.0 ± 27.1 μg/d. OP group: 96.5 ± 33.8 μg/d.	There was no correlation between selenium deficiency and BMD.	-
Rivas et al. 2012 [203]	Cross-sectional study	Spain	280 women ≥ 18 years	24-hr diet recall FFQ	Dietary antioxidants with calcaneous BMD	18–35 yr group: Se: 60.21 μg/d. 36–45 yr group: Se: 69.56 μg/d. >45 yr group: Se: 75.81 μg/d.	Positive association was observed among BMD and selenium. Higher antioxidant intake correlated with high BMD score.	+
P-Z et al. 2012 [204]	Cross-sectional study	Spain	335 postmenopausal women	7-day food records	Se and calcium intake status with Ad-SoS at the phalanges	Mean Se intake: 95.5 μg/d.	Elevated selenium intake negatively affected bone mass Ad-SOS score only in low calcium intake group.	+
Chan et al. 2009 [205]	Cross-sectional study	China	441 women ≥ 20–35 years	5-day food records	Dietary intake with hip, FN, and LS BMD	Honkong Se: 80.6 ± 27.2 μg/d. Beijing Se: 46.4 ± 14.5 μg/d.	Beijing had lower selenium intake. There was no association between selenium and BMD.	-
Sun et al. 2014 [206]	Case–control study	China	726 hip fracture 726 controls	Semi-quantitative FFQ	Dietart intake of antioxidant with hip fracture	Cases: Men: 43.5; Women: 40.8 (μg/d). Controls: Men: 48.3; Women: 47.7 (μg/d).	Higher dietary intake of Se associated with a lower risk of hip fracture. OR of hip fracture: 0.43 (0·26–0·70).	+
Wang, Y. et al. 2019 [207]	Cross-sectional study	China	6267 Adults	Validated semi-quantitative FFQ	BMD at the phalanges	OP: 39.1 ± 31.1 μg/d. Non-OP: 44.0 ± 23.3 μg/d.	Lower levels of dietary Se intake associated with a higher prevalence of osteoporosis. OR of OP: 0.72 (0.55–0.94).	+
Zhang et al. 2021 [208]	Longitudinal study	China	17,150 Adults ≥ 20 years	3-day food records	Self-reported history of fracture	Selenium intake quartile: Q1: 20 ± 5; Q2: 31.8 ± 3; Q3: 42.5 ± 4; Q4: 71.2 ± 45 (μg/d).	Non-linear association between selenium intake and fracture. HRs for fracture Q1: 1.07 (0.86–1.33), Q2: 1 (ref), Q3: 1.25 (1.02–1.53), Q4: 1.33 (1.07–1.65). The U-shape dose–response of the association varies by gender and urbanization level.	+

Abbreviations: BMD: bone mineral density; DXA: dual energy X-ray absorptiometry; FFQ: food frequency questionnaire; LS: lumbar spine; FN: femoral neck; Se: selenium; SePP: selenoprotein P; OP: osteoporosis; FRAX: Fracture Risk Assessment Tool; Ad-SoS: amplitude-dependent speed of sound; + means positive correlation with bone health; - means no significant correlation.

**Table 5 molecules-27-00392-t005:** Selenium supplementation trials in humans with osteoporosis.

Author, Year [Ref]	Study Design	Study Population (Treated/Placebo)	Intervention Se Supplement	Intervention Duration and Follow-Up	Primary Outcomes	Secondary Outcomes	Results	+/-
Walsh et al. 2021 [215]	6-month randomized, double-blind, placebo-controlled trial	120 Postmenopausal women ≥55 years with osteopenia or osteoporosis, UK.	1:1:1 randomized trial; received 200 μg, 50 μg, or placebo per day of sodium selenite.	26 weeks/ 6 month	uNTX to Cr.	Serum Se SePP, BMD, other BTM, muscle function antioxidant, and inflammatory markers.	Urine NTx to creatinine ratio did not differ significantly between treatment groups at 26 weeks. None of the secondary or mechanistic endpoint measurements differed.	-
Perri et al. 2021 [216]	Single-center, randomized, double-blinded, placebo-controlled, multi-arm, parallel clinical trial	490 Healthy adults, 60–74 years. PRECISE trial: (1998–2015), Denmark.	1:1:1:1 Randomized 100, 200, 300 μg or placebo per day of selenium yeast	6 Month/ Mean 15.9 years	Bone turnover markers, osteocalcin, P1NP, carboxy-terminal collagen crosslinks, bone alkaline phosphatase.	Serum Se, TFT, lipid level, antioxidant, and inflammatory markers.	Selenium supplementation reduced P1NP at 6 months, but there were no significant effects on other BTM or after 5 years.	-

Abbreviations: BMD: bone mineral density; DXA: dual energy X-ray absorptiometry; Se: selenium; SePP: selenoprotein P; PINP: procollagen type I *N*-terminal propeptide; OC: osteocalcin; uNTX to Cr: urinary resorption marker *N*-terminal telopeptide of type I collagen to creatine; BTM: bone turnover markers; - means no significant correlation with bone health.

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
