# Peer review of "The Effects of Selenium on Bone Health: From Element to Therapeutics"

_molecules, 2022, doi:10.3390/molecules27020392_

Round 1
Reviewer 1 Report
The review by Yang T et al. deals with a topic of potential interest, that is the role that a trace element could play in bone pathophysiology.
The intent of the authors is undoubtedly commendable, but there are many major points that need to be changed.
- First of all, a narrative review entitled "Selenium: A Trace Element for a Healthy Skeleton - A Narrative Review" published in Endocr Metab Immune Disord Drug Targets. 2021; 21 (4): 577-585 and featured in Pubmed is present in Pubmed, and it is right that it is mentioned in the text and in the references.
- As this is also a narrative review, I recommend that this be specified in the title.
- One of the major problems of this whole review is the "confusion" that reigns regarding the concepts of osteoporosis, bone mass, bone turnover, bone health which sometimes seem to be used as alternatives to each other. Everything is mixed in a chaotic way, without a minimum of distinction and/or integration between them. This is why in many reviews concerning more general aspects of bone pathophysiology, the term skeletal health is used.
- In the introduction, it is important that the authors indicate how and where (Pubmed, Google Scholar ...) they selected the considered works.
- When the authors describe "dietary intake of selenium has been reported to reduce anti-thyroid peroxidase antibodies ... [81]" they should correctly report that other authors have not confirmed this finding, indeed they have had opposite results (Anastasilakis AD et al., Int J Clin Pract, 2012) citing the work.
- In paragraph 3, "Roles of selenoproteins in bone cells", the authors report that selenium deficiency is associated with low bone mass [154], but they should specify that they are male rats.
- It is not clear what exactly the legend of figure 1 is.
- Since studies concerning the effects of thyroid hormones and / or TSH on bone cells, both in animal models and in human studies, are still highly discordant, I suggest eliminating this topic throughout the paper. Moreover, all the studies cited in this regard are put together without any distinction (e.g. in vitro studies, in vivo studies, studies in animal models, studies of genetics in human populations ...).
- In paragraph 4, "Selenium and Bone: Animal Model Studies", the authors introduce the concept of osteomalacia [205], but it would be useful to define what osteomalacia is and how it differs from osteoporosis.
- In the various summary tables of the studies, I would insert a column indicating "outcomes considered".
- I would give more emphasis in this regard of the fact that not in all the studies, the observed results were correlated with the selenium/selenoproteins dosage, nor are they indicated which dosage methods were used in the studies that instead dosed them.
- A serious shortcoming is represented by the total lack of the role of Lrp8/ApoER2 complex, involved in in Selenoprotein P mediated tissu-specific absorption of selenium (Pietschmann N, Rijntjes E, Hoeg A, Stoedter M, Schweizer U, Seemann P, Schomburg L. Metallomics. 2014 May; 6 (5): 1043-9. doi: 10.1039 / c4mt00003j).
- As a general observation, the description of the studies regarding selenium intake levels with bone health is a disordered and too long list, not easy to read, which brings together, without a logical order, studies on bone turnover markers, BMD, fracture risk, although some studies sometimes consider together, in part or all, these parameters. It would be useful, where possible, to try to group clinical studies, observational and non-observational, on human populations by ethnicity evaluated. Furthermore, it must be better specified that most of these studies were performed on samples too small to reach a statistically valid result.
- The whole part about "Novel therapeutics" is redundant, also because most of the studies are on animal models and/or in vitro on cellular models. It is sufficient to make an extreme synthesis, indicating that the research is moving in this direction, because it is moving in this direction, but that we are still at a predominantly "basic" level.
- In paragraph 7 "Conclusions and Future Prospects", the statement that "romosozumab, calcium supplements, bisphosphonates, and selective estrogen receptor modulators are associated with increase in cardiac risk ..." is too strong and not entirely true. Some distinctions are needed, also mentioning opposite studies: Vestergaard P. Acute myocardial infarction and atherosclerosis of the coronary arteries in patients treated with drugs against osteoporosis: calcium in the vessels and not the bones? Calcif Tissue Int. 2012 Jan; 90(1):22–9, Lu P-Y., Hsieh C-F., Tsai Y-W., Huang W-F. Alendronate and raloxifene use related to cardiovascular diseases: differentiation by different dosing regimens of alendronate. Clin Ther. 2011;33(9):1173–9; Kang JH, Keller JJ, Lin HC. Bisphosphonates reduced the risk of acute myocardial infarction: a 2-year follow-up study. Osteoporos Int. 2013 Jan; 24(1):271–7.; Kang JH, Keller JJ, Lin HC. A population-based 2-year follow-up study on the relationship between bisphosphonates and the risk of stroke. Osteoporos Int. 2012 Oct; 23(10):2551–7., Veronese N, Stubbs B, Crepaldi G, et al. Relationship between low bone mineral density and fractures with incident cardiovascular disease: a systematic review and meta-analysis. J Bone Miner Res. 2017 May; 32(5):1126–35.; Chiodini I., Bolland MJ. Calcium supplementation in osteoporosis: useful or harmful? European Journal of Endocrinology (2018) 178, D13–D25, Chung M, Tang AM, Newberry SJ. Calcium Intake and Cardiovascular Disease Risk. Ann Intern Med. 2017 May 2;166(9):686-687. doi: 10.7326/L17-0136.; Harvey NC, D'Angelo S, Paccou J, Curtis EM, Edwards M, Raisi-Estabragh Z, Walker-Bone K, Petersen SE, Cooper C. Calcium and Vitamin D Supplementation Are Not Associated With Risk of Incident Ischemic Cardiac Events or Death: Findings From the UK Biobank Cohort. J Bone Miner Res. 2018 May;33(5):803-811. doi: 10.1002/jbmr.3375; Cy Fixen, Jennifer Tunoa. Romosozumab: a Review of Efficacy, Safety, and Cardiovascular Risk. Curr Osteoporos Rep. 2021 Feb;19(1):15-22. doi: 10.1007/s11914-020-00652-w.
Author Response
We wish to thank the reviewers for his/her helpful review of our manuscript. We are grateful for the suggestions and insightful comments. I have addressed each of their concerns as outlined below. We are submitting the revised manuscript, and highlighted changes within the text with red colour in response to your suggestions.
We apologize sincerely for all those mistakes in our manuscript. It is a great pleasure for us to publish this research work in the Molecules. Our team appreciate your careful comments with detailed explanation.

Reviewer 2 Report
Summary:
In this manuscript, the authors provide a comprehensive review of trace element selenium in bone health, covering the aspects of selenoproteins, metabolic pathways, animal studies, clinical studies and possible novel therapeutics.
Comments:
This manuscript provides a comprehensive literature review of trace element selenium in bone health. However, the following questions should be addressed.
- The statement “However, anti-osteoporotic drugs… are associated with increase in cardiac risk” in the conclusion section is not correct. Indeed, all the current anti-osteoporotic drugs have side effects but not all of them are associated with cardiac risk.
- Figure 1, RANKL is a ligand and it should not be shown as a cell-membrane receptor on the osteoblasts. Please also clarify the yellow circles (near RANK receptor) in the figure legend.
Specific comments:
- Page 1, abstract, change “… acts crucial for” to “… acts crucially for”
- Page 2, please check grammar in the following sentence “Selenoprotein is …, which protects… and are involved in …”
- Page 2, please change “result in certain circumstance” to “result from certain circumstances”
- Page 5, check the format of reference [121],[122,123]. Should be [121-123]
- Page 5, italicize “in vivo”. And please check this through out the context.
- Page 5, check ref format [132]]
- Page 7, should be NFκB rather than NF-B
- Page 7, are the paragraphs “Selenium supplement encourage activity if GPx and TrxR..” and “GPxs, glutathione proxidases;…” part of the caption of Figure 1? If so, please align the format correctly.
- Table 2, make the “+” or “-“ sign in uppercase, such as “Se+”
Author Response

(The authors gave the same response as above.)

Round 2
Reviewer 1 Report
This reviewer appreciates both the answers and changes made by Authors. Now the paper is easier to be read than the original submission.